# Why is the labor epidural rate low and cesarean delivery rate high? A survey of Chinese perinatal care providers

Peishan Zhao[1]☯, Zhenyu Cai[2]☯, Anna Huang[3], Chunyuan Liu[4], Huiling Li[5], Shuwei Yang[6], Ling-Qun Hu ☉[7]*

1 Department of Anesthesiology and Perioperative Medicine, Tufts Medical Center, Boston, Massachusetts, United States of America, 2 Department of Obstetrics and Gynecology, Aviation General Hospital, China Medical University, Beijing, P.R. China, 3 Mailman School of Public Health, Columbia University, New York, New York, United States of America, 4 Department of Anesthesiology, Liangping County People's Hospital, Chongqing, P.R. China, 5 Department of Obstetrics and Gynecology, The People's Hospital, Peking University, Beijing, P.R. China, 6 No Pain Labor & Delivery—Global Health Initiative (China Chapter), ApgarCARE International, Inc., Xi'an, Shaanxi, P.R. China, 7 Department of Anesthesiology, The Ohio State University Wexner Medical Center, Columbus, Ohio, United States of America

☯ These authors contributed equally to this work.
* lingqun.hu@osumc.edu

**Data Availability Statement:** All relevant data are within the manuscript and its Supporting Information files in the S2 Appendix.

## Abstract

### Objective

China has a high cesarean delivery (CD) and low labor epidural analgesia (LEA) rate. This online survey was conducted to explore the reasons behind this phenomenon and potential solutions.

### Methods

A voluntary, anonymous survey was distributed via both WeChat and professional websites for 4 months amongst groups of Chinese perinatal professionals. Data was collected and analyzed using a Chi-square test and presented as percentages of respondents.

### Results

1412 respondents were recorded (43% anesthesiologists, 35% obstetricians, 15.5% midwives or labor and delivery nurses, and 6.5% others), and 1320 respondents were care providers. It was found that 82.7% (1092/1320) of the provider respondents used CD per patient request in fear of lawsuits or yinao/yibao and 63.4% (837/1320) used CD for respecting superstitious culture. The number one reason (noted by 60.2% (795/1320) of all the three specialties) for low LEA use was lack of anesthesia manpower without statistical difference among specialties. The most recommended solution was increasing the anesthesia workforce, proposed by 79.8% (1053/1320) of the three specialties. However, the top solution provided by the two non-anesthesia specialties is different from the one proposed by anesthesiologists. The later (83%, 504/606) suggested increasing the incentive to provide the service is more effective. The answers to questions related to medical knowledge about

**Funding:** The authors received no specific funding for this work.

**Competing interests:** The authors have declared that no competing interests exist.

CD and LEA, and unwillingness of anesthesiologists, parturients and their family members to LEA were similar for the most part, while the opinions regarding low LEA use related to poor experiences and unwillingness of obstetricians and hospital administrators were significantly divided among the three specialties. In the providers' point of view, the unwillingness to LEA from parturient's family members was the most salient (26.1%, 345/1320), which is more than all care providers, hospital administrators, and parturients themselves (16.8%, 222/1320).

## Conclusion

The reasons for high CD rate and low LEA use are multifactorial. The sociological issues (fear of yinao/yibao and superstitious culture) were the top two contributing factors for the high CD rate in China, while lack of anesthesia manpower was the top response for the low LEA use, which contributes to its being the most recommended solution overall from the three specialties. An incentive approach to providers is a short-term solution while training more perinatal care providers (especially among anesthesiologists and midwives), improving billing systems, and reforming legal systems are 3 systemic approaches to tackling this problem in the long-term.

## Introduction

The rate of cesarean delivery (CD) in China was one of the highest in the world at 46.2% in 2007 [1], 28.8% in 2008, and 34.9% in 2014 [2]. Many factors contribute to this high CD rate, with one of the possible reasons due to fear of labor pain [3, 4]. This is strictly independent of cultural factors and exists in up to 68% of Chinese women and 91% of Iranian women [5, 6]. Low rates of labor epidural analgesia (LEA) result in high rates of CDs [7]. Three large impact studies with approximately 50,000 parturients spearheaded by No Pain Labor & Delivery–Global Health Initiative (NPLD-GHI) since its inception in 2008 has confirmed that implementing 24/7 LEA services is correlated with lower rates of CD in 3 different settings (municipal maternal hospitals, academic institutions, and rural area hospitals) [8–10]. Anesthesiologists could play an important role in effectively decreasing the CD rate, given that LEA usage increased from < 1% to ≥ 31% in some hospitals or regions from 2007 to 2018 [8–13].

NPLD-GHI conducted this survey amongst perinatal healthcare providers in China to gauge their opinions of the possible reasons for the high CD rate, low LEA use, and feasible solutions, aiming to decrease unnecessary CD and increase maternal and fetal safety [14]. The null hypothesis was that there is no difference of possible solutions to increase LEA rates and of opinions and knowledge regarding LEA among the various professional providers.

## Materials and methods

This survey does not involve human subjects as defined by the regulations of both the U.S. Department of Health and Human Services and the Food and Drug Administration. No consent was required per Institutional Review Board at Tufts Medical Center. An anonymous online survey questionnaire written in Chinese asked for respondent occupation, locations of work, opinions about LEA and its complications, their concerns, medical compensation, reasons for poor utilization, and suggestions for improvement. The questionnaire was initiated by

the first author "Zhao, P", whose first language is Chinese, and confirmed by the author "Cai, Z", an obstetrician-in-chief, for both linguistic and professional wording. The final set of the survey questions attached as an appendix (S1 Appendix) was validated by Chinese-American healthcare providers from Northwestern University, Harvard Medical School, the Ohio State University, New York University, Howard Medical University, Georgetown Washington University, Mount Sinai at New York, Stanford University, Duke University, and some Chinese institutions. The questionnaire was created at https://www.wenjuan.com/s/zMRbUn4/ and promoted in multiple professional groups run by Jicheng Anesthesia Network (http://www.jcmzw.com) and Chinese Obstetrics and Gynecology Network (http://www.obgy.cn/) (for obstetricians, gynecologists and midwives) via WeChat (the social media app most widely used by Chinese medical professionals to share knowledge and discuss clinical questions) and their official web sites. All surveys were completed based on voluntary and anonymous participation without any incentives or required log-in information.

The questionnaire could be taken by either computers or smartphones but was limited to one per Internet Protocol address to avoid duplicated answers from a single person or device.

The data was converted into an *Excel* spreadsheet for analysis. Results are expressed as numbers and percentages, and a Chi-square analysis was performed using *R* (version 3.4.3, released on November 30, 2017, The *R* Foundation for Statistical Computing, Vienna, Austria) for comparisons with an alpha value of 0.01 reflecting a total number of comparisons more than 20.

## Results

The survey began on September 1st, 2015 and closed on January 1st, 2016. During this period, 4,318 persons accessed the survey from their regions and 1412 of them completed and submitted their responses, with a response rate of 32.7%. The median time to finish the survey was 9 minutes and 51 seconds and 99.2% of the respondents used either Android (65.4%, 923/1412) or iOS devices (33.9%, 478/1412). Only 11 respondents used computers. Respondents included 42.9% (606/1412) anesthesiologists (ANs), 35.1% (495/1412) obstetricians (OBs), 15.5% (219/1412) midwives (11.5%) and labor & delivery nurses (4%) (MNs), 2.3% (33/1412) hospital administrators and 4.2% (59/1412) specialties unspecified. 62.5% of respondents work in public general hospitals, 26.6% in public women's hospitals, and 8.3% in private hospitals with > 90% of respondents working at Level II (municipal) or III (tertiary) hospitals (Level III is the highest level in the Chinese ranking system). The general results above are the same as the first portion of our survey reports [15].

Surveyed subjects represented all 22 provinces, 5 autonomous regions, and 4 municipalities in mainland China. Mean response persons are 45 for these 31 regions, ranging from 1 in Tibet to 138 in Jiangsu province. Based on the Chinese population Census in 2015, the adjusted mean respondent is 9 per region, ranging from 3 in Tibet to 26 in Beijing per 100,000 population. The responses were weighted by population density ranging from 1% to 9% with the average between 1.5% and 5.9%. The correlation between the national population percentage and the survey response rate in 31 individual regions was analyzed and presented linearly in Fig 1 with $R^2 = 0.879$ (regional survey response rate (Y) = 1.8 x regional population percentage of the nation $(X)^{1.2}$).

Fig 2A shows responses to the reasons for the high CD rate; 68.8% (971/1412) of respondents chose more than one answer, demonstrating that the issue is multifactorial. The first two items (Q1-a, b), which were unrelated to medical outcomes seemed to be the most prevalent issues, as confirmed among the three specialties: 1) 82.7% (1155/1320) worried about lawsuits or yinao/yibao (verbal and physical harm toward healthcare providers), and 2) 63.4% (837/

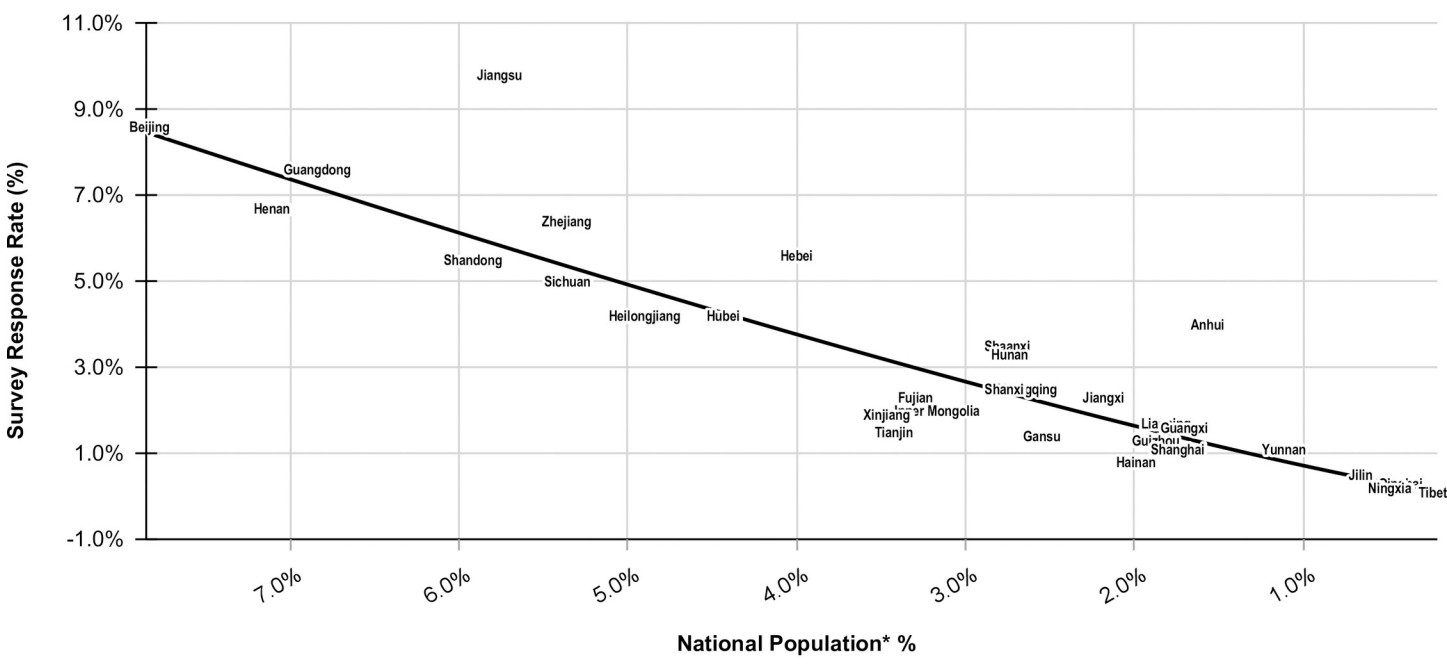

**Fig 1. Correlation between the national population percentages and the survey response rates in 31 regions in mainland China.** * Based on the data from National Bureau of Statistics of China. Annual by Province. 2015. Accessed December 13, 2020. available from: https://data.stats.gov.cn/english/easyquery.htm?cn=E0103.

1320) believed that elective CD was driven by selection of an "auspicious date" from Chinese culture. It seemed that OBs worried more about yinao/yibao whilst ANs felt the culture factor more strongly than other two groups with statistically significant differences. In the following three items relating to medical knowledge, three specialties agreed each other without statistically significant difference (Q1-c to e in Fig 2A, P > 0.01); More non-provider respondents (not shown on Figs) believed that CD is safe for babies (16.3%, 15/92) and mothers (8.7%, 8/ 92) experiencing no long-term complications (4.3%, 4/92), which differed significantly the opinions of healthcare providers. In terms of monetary gain, respondents from the 3 specialties differed (OBs 4% of 495 vs. ANs 20% of 606 vs. NMs 7.8% of 219) regarding the statement, "CD is very safe along with better incentive than VD with LEA" (Q1-f in Fig 2A, p < 0.001). This is consistent with the item Q2c (shown in Fig 2B below) 'More incentive from CD and lost money if providing LEA for VD", which was also indicated predominantly by ANs (OBs 3.8% vs. ANs 19% vs. NMs 5.5%).

Additionally, referring to Fig 2B regarding reasons of low LEA rate, the top 3 responses were: 1) insufficient number of ANs (60.2%, 795/1320, Q2-f), 2) and not enough MNs (31.4%, 415/1320, and Q2-g), and 3) unwillingness to provide LEA because of provider, parturient, and parturients' families' hesitancy. The answers to the first two questions were consistent among the three specialties without statistical significance. Responses regarding unwillingness of different parties to perform LEA overall relate to parturient family members (26.1%, 345/ 1320, Q2-l), ANs (25.6%, 338/1320, Q2-j), and parturients themselves (16.8%, 222/1320, Q2-k). The most responses for this reason came from ANs (44.1% and 18.2% out of 606 respondents), regarding OBs (25.2%, 332/1320, Q2-h) and hospital administrators (11.7%, 154/1320, Q2-i) following. Inexperience (7.3%, 95/1320, Q2-d) or negative experiences (6.2%, 82/1320, Q2-e) also seem to play a role as responded by OBs (9.9%, 49/495). With regards to

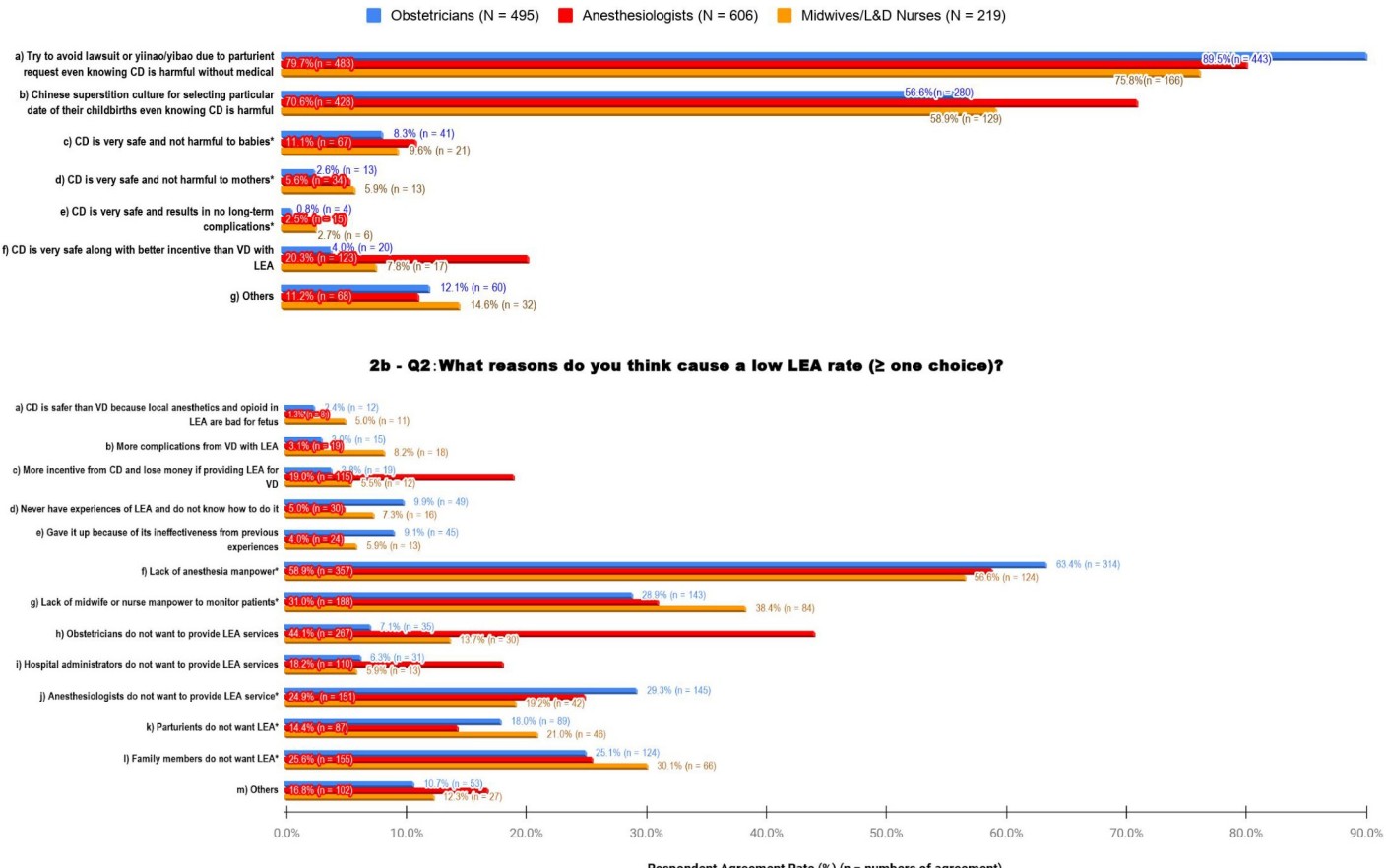

**Fig 2. Reasons of high CD rate and low LEA use.** a—Q1: What are your opinions on reasons for high CD rate? (≥ one choice). b—Q2: What reasons do you think cause a low LEA rate? (≥ one choice). Q1: question 1, Q2: question 2, CD: cesarean delivery, LEA: labor epidural analgesia, N: numbers of the individual study group, n: numbers of the respondents; *: no statistically significant difference at the level of *p*-value = 0.01.

VD with LEA, 6.3% (83/1320) of the survey professionnel believe "CD is better than VD", with 2.3% (31/1320) worried about fetus (Q2-a) and 3.9% (52/1320) worried about mothers (Q2-b). Most of these concerns came from MNs (5.0% and 8.2% out of 219, respectively), which is lower than overall concerns for CD alone (17.1%, Q1-c~e, Fig 2A).

Suggestions to promote LEA usage were examined in Q3 (Fig 3) using a multiple-choice question. The top three suggestions proposed by three specialties were to increase the number of anesthesia personnel (79.8%, 1053/1320), incentive amount (71.1%, 939/1320) and midwife personnel (62.8%, 829/1320). The opinions from the 3 specialties differ by order with statistically significance, i.e. OBs (N = 495) ranking 1) ANs manpower (85%), 2) midwives manpower (68%), and 3) compensation (59%); ANs (N = 606) ranking 1) compensation (83%), 2) ANs manpower (77%), and 3) midwives manpower (52%); and MNs (N = 219) ranking 1) midwives manpower (82%), 2) ANs manpower (76%), and 3) compensation (66%). Overall, increasing AN manpower was the top response consistent with the number one reason for low LEA rate (Q2 f, Fig 2B).

## Discussion

The reasons for the high rate of CD and low rate of LEA in China are complex and multifaceted. This survey attempted to gather opinions from perinatal care providers on these issues

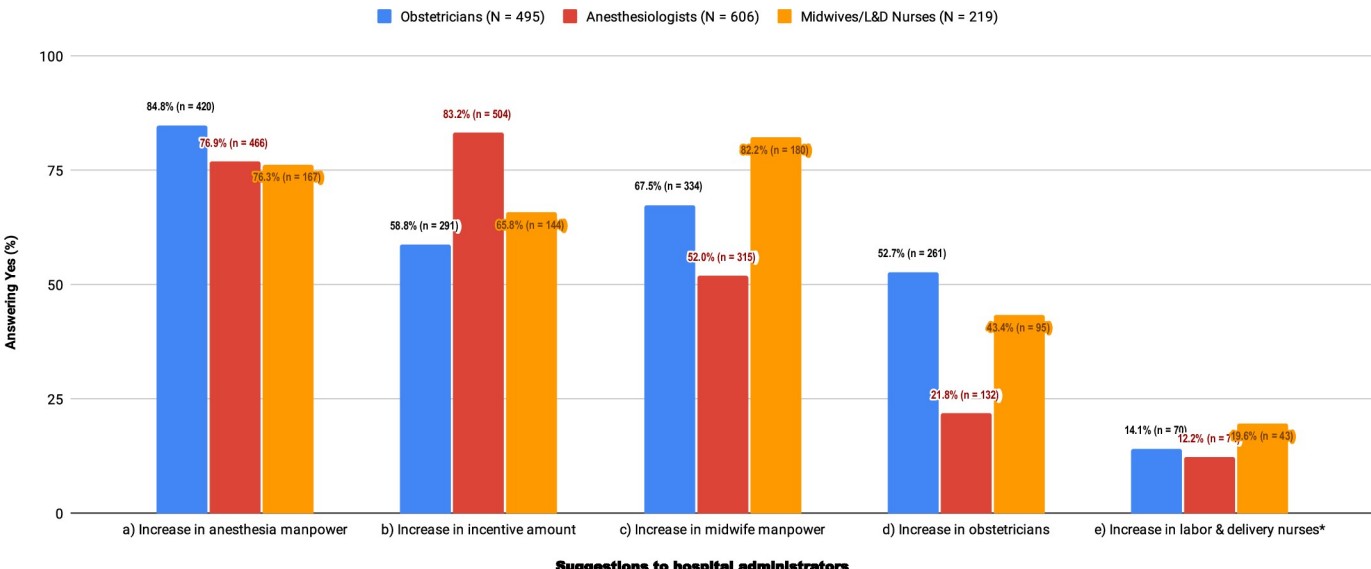

**Fig 3. Q3: What to suggest hospital administrators to promote labor epidural analgesia service presuming they are willing to support?** (≥ one choice). Q3: question 3, N: numbers of the individual study group, n: numbers of the respondents; *: no statistically significant difference at the level of *p*-value = 0.01.

and seek potential solutions. Consistently, three specialties agreed that the top 2 reasons for the high rate of CD are fear of consequences from malpractice and superstition culture, and the top reason for the low rate of LEA is lack of ANs manpower, which was also the top suggested solution to increase LEA and subsequently decrease CD rate. Most notably, more ANs responded that incentives regarding not only CD (Q1-f) but also VD with LEA (Q2-c) were important in a short-term solution to increase use of LEA. Although the solutions seem simple at a glance, the implementation requires careful planning and great efforts. All three specialties agreed that ANs manpower issue, unwillingness from parturients, their family members, and ANs to utilize LEA, as well as insufficient medical knowledge regarding harmfulness of CD to mothers and babies were possible causes for low LEA rates. However, opinions in the remaining question items were divided.

Because of the "One-Child Policy", many Chinese families unrealistically expect a "perfect" baby with no complications and often file lawsuits against doctors in the case of adverse events during childbirth. Chinese medical malpractice claims are reviewed in the criminal court, with the jury often associating malpractice with intentional harm and granting large sums of compensation to the victims. This practice discourages physicians from performing procedures that may go against the wishes of the patient and their families. In more extreme cases, patients/families may even perform yinao/yibao against physicians when faced with adverse outcomes [16]. Although the reasons for yinao/yibao are complex, the main purpose is essentially for patients and their families to obtain financial compensation in out-of-court settlement from hospitals and doctors in the case of adverse clinical outcomes [16]. Yinao/yibao is a large factor preventing Chinese physicians from making appropriate medical decisions [16]. Family involvement in consent for procedures in the Chinese medical-legal system is another hurdle to not only providing LEA but also for physicians in making appropriate clinical decisions for the parturient. Most Chinese parturients could not receive LEA without their family members' consent, and in all 3 groups surveyed, a total 26.1% of providers agreed that LEA was not provided because of non-consenting family members. In this respect, the medical-legal system needs major reform to provide healthcare in the best interest for parturients, their family members, and physicians alike.

Superstitious culture surrounding the birth of the child also affects the usage of LEA and CD. Many Chinese parents/grandparents believe their child would be smarter or wealthier if the baby is delivered at a specific date and time frame, and CD is the only way to meet their wishes. This "superstitious culture" affects Chinese CD rate dramatically, with estimations of this phenomenon contributing up to 27% of coded "social reasons" in one study [17]. A total of 7 studies were collected in a systematic review reporting incidence of this phenomenon ranging from 1.1% to 19.2% [18]. Auspicious birth dates as a reason for CD has been popularized not only in China but also in Thailand and in the United States [19–21]. Without proper LEA during vaginal delivery and compensation for longer hours of bedside care, the rise in popularity in opting for CD comes as no surprise. For parturients, CD is quicker and less painful than vaginal delivery (discounting postoperative period), and for providers (assuming no complications), the procedure is much better compensated. Thus, in past years CD has become the simplest and most socially acceptable method of delivery [22]. The misconceptions surrounding the benefit of CD spread among parturients, their family members, and even providers themselves have been seen in this survey in spite of the evidence supporting the benefits of VD including fewer complications for both mother and infant.

The top solution (increasing AN manpower) agreed upon by the three specialties, is consistent with previous literature [23]. However, it is important to note that among ANs "increasing incentives" was the most popular option. These choices reflect the demographic of physicians In China where there are currently only 5 anesthesiologists per 100,000 people compared to 20 per 100,000 in the US [23–25]. Obstetric anesthesiologists playing a vital role in parturient safety [26], however, the specialty does not officially exist in China. Lack of either anesthesiologists or obstetric anesthesiologists in low/middle income countries including China could be one of the contributors to the higher maternal mortality rate compared to other countries [27]. Increasing the number of anesthesiologists and obstetric anesthesiologists in the largest population in the world offers a long-term solution, but cannot occur instantaneously. With a well-established training system in the United States, training an obstetric anesthesiologist takes at least 5 years of residency and fellowship, given there are no issues of medical student shortage and adequate training spots.

In the short term, the top solution recommended by ANs to increase incentives is unopposed by other specialists. This is especially resounding because the salary of healthcare professionals in China is disproportionately low compared to other countries with more than 50% of compensation driven by case volume [23, 28], similar to the US's fee-for-service system [24, 25]. Studies indicate that increasing incentives to providers rapidly increases LEA and decreases CD rates with better maternal and infant outcomes and increased hospital revenue [8–11, 29]. This might serve as a pathway for a short-term solution while more AN and OB personnel are trained. When presenting preliminary results to participating hospitals administrators, they were surprised at the results and subsequently may review the financial impact of LEA services in Chinese public hospitals [29]. Regarding the personnel who are most responsible for not providing LEA, 44.1% of ANs responded that OBs were unwilling to provide LEA and 29.3% of OBs responded that ANs were unwilling to provide LEA. These responses, however, precludes any outside pressure from hospital administrators, who may also influence their service decision-making. The hospital administration's unwillingness is an underestimated issue nationwide, although this may be skewed because most of the survey respondents work in the hospitals where LEA services are available [8–11, 14, 29].

Although web-based surveying is becoming increasingly popular in medical research as internet technology advances, this survey is limited and may be biased by the lower response rate (32.7% or 1412/4318) as a result of limited internet access and clustered sampling. However, compared to traditional survey methodology, this survey method was still able to reach

more people because of ease of smartphone operation and widespread usage, which could have contributed to the greater than average response rate as reported in a meta-analysis [30]. 99.2% of the respondents used smartphones, which may be indicative of a younger sample population with over 90% of them working in municipal or tertiary hospitals without demographic control. Sampling bias is a limitation to the study, although the rate of the survey respondents did correlate with the regional population proportionally. Additionally, substantial differences in group sizes are a source of sampling and response/nonresponse bias. Although we have intentionally organized the orders of the answers to the three survey questions with certain confirmation to each other, there still may be question order bias as well. Therefore, the results need to be interpreted with cautions.

The most significant findings from this survey are the possible avenues in which short-term and long-term solutions to increase LEA and safely decrease CD rates with better maternal and baby outcomes in China. Given the increase in high-risk parturients (older or/and s/p previously CD) as a result of ending the One Child Policy, CD rates in China may remain high or even increase greater than they have been historically. Providing safe and effective LEA and obstetric anesthesia care is even more crucial now than ever. Inadequate knowledge, experiences, and patient awareness were reflected from the results in this survey and should be addressed urgently [15].

## Conclusions

Our survey revealed that the main speculated reasons for the high rate of CD and low rate of LEA in China as surveyed from the opinions of OBs, ANs, and MNs across China are a fear of consequences from medical malpractice and superstition and insufficient patient awareness and physician knowledge of LEA and CD compounded with issues in family member and patient consent. Unbalanced financial incentives between CD and VD with/without LEA, and lack of healthcare providers overall were also the issues addressed in the survey. With the advocacy of the Chinese central government and support from local hospital administration, revising hospital reimbursement, reforming medical-legal systems, and training more ANs and MNs would be the systematic and long-term solutions for these issues. It is feasible and realistic to increase the access of LEA to parturients by incentivizing obstetric anesthesia services in the L&D floors in the short-term.

With hope, this survey results could provide valuable insight for clinical service and health policy decision-making in hospitals at the regional and national levels in China and other lower to middle-income countries.

## Supporting information

**S1 Appendix. The survey questions in both the original language (Chinese) and English.** (PDF)

**S2 Appendix. The raw survey data for both Part I and Part II (for this manuscript).** (XLSX)

## Acknowledgments

The authors would like to thank all the individual participants of No Pain Labor & Delivery— Global Health Initiative since 2008 for continuously supporting. It would be impossible to make such definitive improvements of maternal and baby outcomes in this 20% world population without your close to 1000 individual visits.

## Author Contributions

**Conceptualization:** Peishan Zhao, Ling-Qun Hu.

**Data curation:** Peishan Zhao, Anna Huang, Huiling Li, Shuwei Yang, Ling-Qun Hu.

**Formal analysis:** Anna Huang, Ling-Qun Hu.

**Funding acquisition:** Zhenyu Cai.

**Investigation:** Peishan Zhao, Ling-Qun Hu.

**Methodology:** Peishan Zhao, Anna Huang, Shuwei Yang, Ling-Qun Hu.

**Project administration:** Zhenyu Cai, Chunyuan Liu, Shuwei Yang, Ling-Qun Hu.

**Resources:** Zhenyu Cai, Chunyuan Liu, Huiling Li, Shuwei Yang, Ling-Qun Hu.

**Software:** Shuwei Yang.

**Supervision:** Peishan Zhao, Zhenyu Cai, Ling-Qun Hu.

**Validation:** Peishan Zhao, Zhenyu Cai, Chunyuan Liu, Huiling Li, Ling-Qun Hu.

**Visualization:** Chunyuan Liu, Huiling Li, Shuwei Yang, Ling-Qun Hu.

**Writing – original draft:** Peishan Zhao, Ling-Qun Hu.

**Writing – review & editing:** Peishan Zhao, Zhenyu Cai, Anna Huang, Ling-Qun Hu.

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
