## [Decision Letter · Decision Letter 0]

2 Feb 2021

PONE-D-21-00405

Why is the labor epidural rate low and cesarean delivery rate high? A Survey of Chinese Perinatal Care Providers

PLOS ONE

Dear Dr. Hu,

Thank you for submitting your manuscript to PLOS ONE. After careful consideration, we feel that it has merit but does not fully meet PLOS ONE’s publication criteria as it currently stands. Therefore, we invite you to submit a revised version of the manuscript that addresses the points raised during the review process.

We look forward to receiving your revised manuscript.

Kind regards,

Huafeng Wei, MD, PhD

Academic Editor

PLOS ONE

2. Please include additional information regarding the survey or questionnaire used in the study and ensure that you have provided sufficient details that others could replicate the analyses. For instance, if you developed a questionnaire as part of this study and it is not under a copyright more restrictive than CC-BY, please include a copy, in both the original language and English, as Supporting Information.  If the original language is written in non-Latin characters, for example Amharic, Chinese, or Korean, please use a file format that ensures these characters are visible.

3. In your Methods section, please include additional information about your dataset and ensure that you have included a statement specifying whether the collection method complied with the terms and conditions for the application WeChat.

4. Please state whether you validated the questionnaire prior to testing on study participants. Please provide details regarding the validation group within the methods section.

5. Please include your tables as part of your main manuscript and remove the individual files. Please note that supplementary tables (should remain/ be uploaded) as separate "supporting information" files.

6. Thank you for stating the following in the Competing Interests section:

We note that one or more of the authors are employed by a commercial company: ApgarCARE International, Inc.

(2) Please also provide an updated Competing Interests Statement declaring this commercial affiliation along with any other relevant declarations relating to employment, consultancy, patents, products in development, or marketed products, etc.  

7. We note that Figure 1 in your submission contains map images which may be copyrighted. All PLOS content is published under the Creative Commons Attribution License (CC BY 4.0), which means that the manuscript, images, and Supporting Information files will be freely available online, and any third party is permitted to access, download, copy, distribute, and use these materials in any way, even commercially, with proper attribution. For these reasons, we cannot publish previously copyrighted maps or satellite images created using proprietary data, such as Google software (Google Maps, Street View, and Earth). For more information, see our copyright guidelines: http://journals.plos.org/plosone/s/licenses-and-copyright.

(1) You may seek permission from the original copyright holder of Figure 1 to publish the content specifically under the CC BY 4.0 license. 

Reviewers' comments:

Reviewer's Responses to Questions

**Comments to the Author**

1. Is the manuscript technically sound, and do the data support the conclusions?

Reviewer #1: Yes

Reviewer #2: Yes

2. Has the statistical analysis been performed appropriately and rigorously? 

Reviewer #1: Yes

Reviewer #2: Yes

3. Have the authors made all data underlying the findings in their manuscript fully available?

Reviewer #1: Yes

Reviewer #2: Yes

4. Is the manuscript presented in an intelligible fashion and written in standard English?

Reviewer #1: Yes

Reviewer #2: Yes

5. Review Comments to the Author

Reviewer #1: Dear Editor,

I am honored to be invited to review this article.

As an obstetrician for more than 16 years, I do realize that cesarean delivery rates

overall are very different. The differences in rates depend on not only medical indications but

also social-economic status. We do have a lot of clinical studies addressing unnecessary

cesarean deliveries medically. Unfortunately, the vast majority of cesarean deliveries occur far

beyond our majorities of research. It is fortunate that the authors picked up this topic in this very

unique view. The evidence that the authors provided have convinced me that providing epidural

labor analgesia can safely reduce the cesarean delivery rate. This conclusion was drawn not

only from their own large impact studies in different levels of hospitals but also from RCT studies

reviewed by the Cochrane group in 2018. In other words, the background is solid and real.

As I read the manuscript, I also realized that the unnecessary cesarean deliveries in

China have equaled the total amount of cesarean deliveries annually in the United States. I felt

that this topic and approach would also be very beneficial to the medical societies in the rest of

the world other than China. The high rates of cesarean delivery are epidemic if not pandemic

epidemiologically since the unnecessary cesarean deliveries have clearly causal effects on

maternal and infant morbidity and mortality. There are some unique cultural aspects, such as

superstitious culture which was only applied to the local community. However, such a situation

occurs in about 20% of the world population and maybe even more in other Asian countries

around.

Having talked about many positive aspects of this manuscript, I would like to make a few

comments and recommendations:

Major issues:

1) Table 1 and Table 2 could be presented better. The authors may use histogram for

better and more clear view.

2) Web-based survey is a relatively new form with a lower respondent rate than traditional

surveys via email, mail, or interview. The authors may analyze distribution bias, and

discuss related biases to help readers to interpret data objectively.

Minimal issues:

1) If the authors keep Table 1 and Table 2, two columns for each specialty should become

one in the format “xxx (%)” in order to simplify the tables..

2) It may be better to list the survey questions as an appendix at the end.

I am so impressed by the feasible solution concluded in the study. I am very glad that

you accepted this manuscript and started its reviewing proces

Reviewer #2: This is a manuscript reporting the results of a provider survey in China to assess the causes of high cesarean delivery and low labor epidural analgesia rates. The authors found that China has a high cesarean rate due to cultural and legal considerations, while the low epidural analgesia usage is impacted by lack of providers, incentives and social issues. The survey explored some methods to reduce the cesarean rate and increase the usage of epidural analgesia. While these are complex issues that, naturally, cannot be explained in a single survey, the information being reported is of interest for the health care system of the World’s largest population. The manuscript could use a little improvement in organization and language editing, as the tenses fluctuate even within a sentence.

Major critiques

Data collected via Wechat: while this is the predominant communications tool in China, can the authors estimate the penetrance of the survey – in other words, how many of each providers received the survey, not just those that opened it but did not answer.

Abstract is not fully descriptive of the results

Please do not repeat data that exists in the tables and figures in the Results section. You should describe what the significant findings were, but the multiple numbers are confusing and require the reader to do a lot of work to comprehend.

To some extent, the survey is designed to create the result that the authors hypothesized. It is challenging to understand how the reported numbers do – or do not – support or refute the narrative. For example, “3.5-5 times as many ANs (19%) as OBs (3.8%) and MNs (5.5%) responded that the low LEA rate is related to a smaller financial risk related to CD and more risk of monetary loss for LEA (Q2-c, p < 0.001), consistent with item Q1-f, which reveals that 20% ANs, 4.0% OBs and 7.8% MNs believed that monetary gain is a possible cause for the high CD rate observed.” Does this mean that only 20% of anesthesia providers and 4% of obstetric providers believe financial incentives lead to the outcome? This seems like a small minority support the opinion being put forth. Do 80% of anesthesia and 96% of obstetric providers believe otherwise?

Minor comments

Yinao/yinbao is written three different ways throughout the document. Please use only 1 version

Introduction

1st paragraph:

Many factors likely contributed to this high CD rate, with one of the possible reasons being fear of labor pain

The fear of labor pain is independent of culture…

Low rates of labor epidural analgesia (LEA) might result in high rates

“In fact, with countermeasures and education in place,” this is speculation, unless the authors have evidence they can quote. Please state the facts as known (LEA has increased in some regions) without hypothesizing why

Methods

page 4: All questionnaires could be taken by either computers or smartphones based on their own choice (it’s not really a clinical practice to answer a survey)

9'51'' – change to minutes & seconds

Results

Page 5: Three professionals (OBs 4% vs. ANs 20% vs. NMs 7.8%) disagreed… – this might be better to report the numbers rather than only %

Discussion

Page 7: “This survey attempted to gather opinions from perinatal care providers on these issues and seek potential solutions”

…for the high rate of CD were fear of consequences…

Page 8

This "superstition culture" affects Chinese CD rate dramatically contributing up to 27% of coded “social reasons” in one study or 21% of coded “no indication” in another study. (only one study is referenced, please add both)

Obstetric anesthesiologists, the physicians play a vital role in parturient safety are also lack in China, which may have resulted in higher maternal mortality in low/middle income countries. – I don’t know what this sentence is trying to say

Page 9

Studies indicate that increasing incentives to providers rapidly increases LEA and decreases CD rates with better maternal and infant outcomes, while the hospital revenue increases. This might serve as a short-term solution while more AN and OB personnel are trained.13, 23 (which may repeat the American history that occurred twice.24,25 – this is not necessary)

Page 10

Although modifying the national medicolegal system, billing codes…

Delete “This survey will guide NPLD-GHI in its goals, provide information on clinical practice and policy making in China and other lower to middle income countries.” That is not appropriate for a journal article, but is a statement of internal strategy

Figure 1: the donut chart should be an itemized list, as the circlular format is not relevant, and the colors are not applied to the left half of the figure.

Table 1: it might be clearer if you place the percentages inside of the column (e.g. 360 (26%) )

6. PLOS authors have the option to publish the peer review history of their article (what does this mean?). If published, this will include your full peer review and any attached files.

Reviewer #1: **Yes: **Qing Ding, MD, FACOG

Reviewer #2: No

---

## [Author Response · Author response to Decision Letter 0]

7 Mar 2021

Please see the attached file titled "Respond to Reviewers".

---

## [Decision Letter · Decision Letter 1]

8 Apr 2021

PONE-D-21-00405R1

Why is the labor epidural rate low and cesarean delivery rate high? A survey of Chinese perinatal care providers

PLOS ONE

Dear Dr. Hu, 

Thank you for submitting your manuscript to PLOS ONE. After careful consideration, we feel that it has merit but does not fully meet PLOS ONE’s publication criteria as it currently stands. Therefore, we invite you to submit a revised version of the manuscript that addresses the points raised during the review process.

We look forward to receiving your revised manuscript.

Kind regards,

Huafeng Wei, MD, PhD

Academic Editor

PLOS ONE

Journal Requirements:

Additional Editor Comments (if provided):

Please address the comments from reviewer 2.

Reviewers' comments:

Reviewer's Responses to Questions

**Comments to the Author**

1. If the authors have adequately addressed your comments raised in a previous round of review and you feel that this manuscript is now acceptable for publication, you may indicate that here to bypass the “Comments to the Author” section, enter your conflict of interest statement in the “Confidential to Editor” section, and submit your "Accept" recommendation.

Reviewer #1: All comments have been addressed

Reviewer #2: All comments have been addressed

2. Is the manuscript technically sound, and do the data support the conclusions?

Reviewer #1: Yes

Reviewer #2: Yes

3. Has the statistical analysis been performed appropriately and rigorously? 

Reviewer #1: Yes

Reviewer #2: Yes

4. Have the authors made all data underlying the findings in their manuscript fully available?

Reviewer #1: Yes

Reviewer #2: No

5. Is the manuscript presented in an intelligible fashion and written in standard English?

Reviewer #1: Yes

Reviewer #2: Yes

6. Review Comments to the Author

Reviewer #1: I have no additional comments at this time. The authors have adequately addressed your comments raised in a previous round of review and you feel that this manuscript is now acceptable for publication. Thank you for having me review the manuscript. Good luck for publishing your work. Thanks.

Reviewer #2: Line 48: ‘without statistical difference among specialties.’

Also, delete: ‘It was consistent with’ and make: the most recommended solution was increasing

Line 50: This is a confusing sentence, as it is not clear which group relates to the numbers. Please revise to something like: However, the top solution provided by the two non-anesthesia specialties was different from the one proposed by anesthesiologists. The later (83%, 504/606) suggested increasing the incentive to provide the service is more effective.

Line 95: there are two commas

Line 124: percentages, please

Line 261: fewer complications

Line 263: Top solution (top one is redundant)

Line 268: however, the specialty does not officially exist (I am assuming the people officially exist!)

Line 286: delete ‘seriously’

Line 295: increasingly popular (‘more’ is redundant)

7. PLOS authors have the option to publish the peer review history of their article (what does this mean?). If published, this will include your full peer review and any attached files.

Reviewer #1: No

Reviewer #2: No

---

## [Author Response · Author response to Decision Letter 1]

21 Apr 2021

The designated file has been uploaded.

---

## [Editor Report · Decision Letter 2]

26 Apr 2021

Why is the labor epidural rate low and cesarean delivery rate high? A survey of Chinese perinatal care providers

PONE-D-21-00405R2

Dear Dr. Hu,

We’re pleased to inform you that your manuscript has been judged scientifically suitable for publication and will be formally accepted for publication once it meets all outstanding technical requirements.

Kind regards,

Huafeng Wei, MD, PhD

Academic Editor

PLOS ONE

---

## [Editor Report · Acceptance letter]

14 May 2021

PONE-D-21-00405R2 

Why is the labor epidural rate low and cesarean delivery rate high? A survey of Chinese perinatal care providers 

Dear Dr. Hu:

I'm pleased to inform you that your manuscript has been deemed suitable for publication in PLOS ONE. Congratulations! Your manuscript is now with our production department. 

Kind regards, 

on behalf of

Dr. Huafeng Wei 

Academic Editor

PLOS ONE